# Development and validation of automated computer-aided risk scores to predict in-hospital mortality for emergency medical admissions with COVID-19: a retrospective cohort development and validation study

Muhammad Faisal [1,2,3] Mohammed Mohammed,[1,4] Donald Richardson,[5] Massimo Fiori,[6] Kevin Beatson[6]

For numbered affiliations see end of article.

**Correspondence to**
Dr Mohammed Mohammed;
m.a.mohammed5@bradford.ac.uk

## ABSTRACT

**Objectives** There are no established mortality risk equations specifically for unplanned emergency medical admissions which include patients with SARS-19 (COVID-19). We aim to develop and validate a computer-aided risk score (CARMc19) for predicting mortality risk by combining COVID-19 status, the first electronically recorded blood test results and the National Early Warning Score (NEWS2).

**Design** Logistic regression model development and validation study.

**Setting** Two acute hospitals (York Hospital—model development data; Scarborough Hospital—external validation data).

**Participants** Adult (aged ≥16 years) medical admissions discharged over a 24-month period with electronic NEWS and blood test results recorded on admission. We used logistic regression modelling to predict the risk of in-hospital mortality using two models: (1) CARMc19_N: age+sex+NEWS2 including subcomponents+COVID19; (2) CARMc19_NB: CARMc19_N in conjunction with seven blood test results and acute kidney injury score. Model performance was evaluated according to discrimination (c-statistic), calibration (graphically) and clinical usefulness at NEWS2 thresholds of 4+, 5+, 6+.

**Results** The risk of in-hospital mortality following emergency medical admission was similar in development and validation datasets (8.4% vs 8.2%). The c-statistics for predicting mortality for CARMc19_NB is better than CARMc19_N in the validation dataset (CARMc19_NB=0.88 (95% CI 0.86 to 0.90) vs CARMc19_N=0.86 (95% CI 0.83 to 0.88)). Both models had good calibration (CARMc19_NB=1.01 (95% CI 0.88 to 1.14) and CARMc19_N:0.95 (95% CI 0.83 to 1.06)). At all NEWS2 thresholds (4+, 5+, 6+) model, CARMc19_NB had better sensitivity and similar specificity.

**Conclusions** We have developed a validated CARMc19 scores with good performance characteristics for predicting the risk of in-hospital mortality. Since the CARMc19 scores place no additional data collection burden on clinicians, it may now be carefully introduced

## STRENGTHS AND LIMITATIONS OF THIS STUDY

⇒ This study provides a computer-aided risk of in-hospital mortality for unplanned admissions with COVID-19 using National Early Warning Score (NEWS2) and routine blood test results.

⇒ About 20%–30% of admissions do not have both NEWS2 and blood test results and so we have developed two scores (computer-aided risk score (CARMc19)_N and CARMc19_NB) reflecting those with/without blood test results.

⇒ Patients with COVID-19 were determined by COVID-19 swab test results (hospital or community) and clinical judgement and so our findings are constrained by the accuracy of these methods.

⇒ Our two hospitals are part of the same NHS Trust and this may undermine the generalisability of our findings, which merit further external validation.

⇒ CARMc19 scores place no additional data collection burden on clinicians and are readily automated.

and evaluated in hospitals with sufficient informatics infrastructure.

## INTRODUCTION

The SARS-19 produced 'COVID-19' infection in individuals with symptoms that has challenged healthcare systems globally (Coronaviridae Study Group of the International Committee on Taxonomy of Viruses[1]). Patients with COVID-19 admitted to the hospital during the early stages of the pandemic were at severe risk of developing the severe disease with life-threatening respiratory and/or multiorgan failure[2,3] with a high risk of mortality.

Early diagnosis and management of patients with COVID-19 was key in providing high-quality care, which included palliative care,

isolation and escalation to critical care. Early Warning Scores (EWS) are commonly used in hospitals worldwide,[4] and in the National Health Service (NHS) hospitals in England, the patient's National Early Warning Score (NEWS) is used to identify patients at risk of deterioration.[5] We have developed two automated risk equations to predict the patient's risk of in-hospital mortality (CARM_N and CARM_NB) using NEWS only (CARM_N)[6] and NEWS+blood test results (CARM_NB)[7] following emergency medical admission to hospital. We found CARM_NB performed similar to consultant clinicians.[8]

NEWS2 was published in December 2017 as an update to NEWS[4] that considered new confusion or delirium and allocated three points (the maximum for a single variable). NEWS2 also offers two scales for oxygen saturation (scale 1 and scale 2). Scale 2 is used for patients at risk of hypercapnic respiratory failure who have a lower oxygen saturation target of 88%–92%.

While hospitals continued to use NEWS2 during the COVID-19 pandemic, little was known at the time about how NEWS2 and CARM scores perform in monitoring patients with COVID-19. In this study, we aimed to develop and validate an automated computer-aided risk score (CARMc19) using on admission NEWS2 and blood test results for predicting mortality in our patient cohort that included a large number with a diagnosis of COVID-19. This approach is clinically useful because it places no additional data collection burden on staff for monitoring patients with COVID-19. It must be stressed that this algorithm was developed at a time that predated widespread vaccination and the development of other evidence-based treatments for COVID-19 disease. The Randomised Evaluation of COIVD-19 Therapy (RECOVERY) study was ongoing in the trust during the development of this algorithm.[9]

## METHODS
### Setting and data
Our cohorts of emergency medical admissions are from two acute hospitals which are approximately 65 km apart in the Yorkshire and Humberside region of England—Scarborough Hospital (SH) (n~300 beds) and York Hospital (YH) (n~700 beds), managed by York Teaching Hospitals NHS Foundation Trust. We selected these hospitals because they had electronic NEWS2, collected as part of the patient's process of care since April 2019, and were agreeable to the study.

We considered all consecutive adult (aged ≥18 years) non-elective or emergency medical admissions discharged over a course of 3 months (11 March 2020 to 13 June 2020) with electronic NEWS2. For each emergency admission, we obtained a pseudonymised patient identifier, patient's age (years), sex (male/female), discharge status (alive/dead), admission and discharge date and time, diagnoses codes based on the 10th revision of the International Statistical Classification of

Diseases (ICD-10), NEWS2 (including its subcomponents respiratory rate, temperature, systolic pressure, pulse rate, oxygen saturation, oxygen supplementation, oxygen scales 1 and 2 and alertness including confusion), blood test results (albumin, creatinine, haemoglobin, potassium, sodium, urea and white cell count) and Acute Kidney Injury (AKI) score.

The diastolic blood pressure was recorded at the same time as systolic blood pressure. Historically, diastolic blood pressure has always been a routinely collected physiological variable on vital sign charts and is still collected where electronic observations are in place. NEWS2 produces integer values that range from 0 (indicating the lowest severity of illness) to 20 (the maximum NEWS2 value possible) (online supplemental appendix table S1). The index NEWS2 was defined as the first electronically recorded NEWS2 within ±24 hours of the admission time. We excluded records where the index NEWS2 (or blood test results) was not within ±24 hours (±96 hours) or was missing/not recorded at all (online supplemental appendix table S2). The ICD-10 code 'U071' was used to identify records with COVID-19. We searched primary and secondary ICD-10 codes for 'U071' for identifying COVID-19.

### Statistical modelling
We began with exploratory analyses including box plots and line plots to show the relationship between covariates and risk of in-hospital mortality. We developed two logistic regression models, known as CARMc19_N and CARMc19_NB, to predict the risk of in-hospital mortality with following covariates: (1) model CARMc19_N uses age+sex+COVID-19 (yes/no)+NEWS2 including subcomponents; (2) model CARMc19_NB extends model CARMc19_N with all seven blood test results and AKI score. The primary rationale for using these variables is that they are routinely collected as part of process of care and their inclusion in our statistical models is on clinical grounds as opposed to the statistical significance of any given covariate.

We used the *qladder* function (Stata[10]), which displays the quantiles of a transformed variable against the quantiles of a normal distribution according to the ladder powers $\left(x^3, x^2, x^1, x, \sqrt{x}, \log(x), x^{-1}, x^{-2}, x^{-3}\right)$ for each continuous covariate and chose the following transformations: (creatinine)$-1/2$, $\log_e(\text{potassium})$, $\log_e(\text{white cell count})$, $\log_e(\text{urea})$, $\log_e(\text{respiratory rate})$, $\log_e(\text{pulse rate})$, $\log_e(\text{systolic blood pressure})$ and $\log_e(\text{diastolic blood pressure})$. We used an automated approach to search for all two-way interactions and incorporated those interactions which were statistically significant (p<0.001) from the MASS library[11] in R.[12]

We developed both models using YH data (development dataset) and externally validated their performance on SH data (validation dataset). The hospitals are part of the same NHS Trust but are geographically separated by about 65 km (40 miles).

**Table 1** Characteristics of emergency medical admissions in development and validation datasets

| Characteristic | Development dataset (YH) | Validation dataset (SH) | Degree of freedom (df) | P value |
|---|---|---|---|---|
| N | 3924 | 2520 | | – |
| Male (%) | 2010 (51.2) | 1247 (49.5) | 1 | 0.181 |
| Mean age (years) (SD) | 67.4 (18.7) | 69.6 (18.9) | 5320 | <0.001 |
| Median length of stay (days) (IQR) | 3.0 (5.8) | 3.7 (6.1) | – | <0.001 |
| COVID-19 (%) | 343 (8.7) | 277 (11.0) | 1 | 0.003 |
| Mortality | | | | |
| Mortality within 24 hours (%) | 30 (0.8) | 32 (1.3) | 1 | 0.058 |
| Mortality within 48 hours (%) | 61 (1.6) | 48 (1.9) | 1 | 0.335 |
| Mortality within 72 hours (%) | 96 (2.4) | 68 (2.7) | 1 | 0.585 |
| In-hospital mortality | 323 (8.2) | 212 (8.4) | 1 | 0.833 |
| Mean NEWS2 (SD) | 2.8 (2.8) | 3.2 (2.8) | 5446 | <0.001 |
| Vital signs | | | | |
| Mean respiratory rate (bpm) (SD) | 19.8 (5.1) | 20.7 (5.6) | 5027 | <0.001 |
| Mean temperature (°C) (SD) | 36.4 (0.9) | 36.3 (1) | 4817 | 0.001 |
| Mean systolic pressure (mm Hg) (SD) | 141.8 (29.2) | 142 (28.5) | 5455 | 0.839 |
| Mean diastolic pressure (mm Hg) (SD) | 79.2 (16.5) | 79 (17.3) | 5193 | 0.545 |
| Mean pulse rate (bpm) (SD) | 89.1 (22.3) | 88.5 (22.1) | 5406 | 0.336 |
| Mean oxygen saturation (SD) | 96.3 (3.1) | 96.1 (3.2) | 5182 | 0.059 |
| Oxygen supplementation (%) | 512 (13) | 362 (14.4) | 1 | 0.142 |
| Mean oxygen flow rate (units) (SD) | 7.1 (5.7) | 6.1 (5.3) | 811 | 0.007 |
| Oxygen scale 2 (yes) (%) | 240 (6.1) | 163 (6.5) | 1 | 0.605 |
| Alertness | | | | |
| Alert (%) | 3510 (89.4) | 2243 (89) | 5 | 0.010 |
| Baseline confusion (%) | 27 (0.7) | 23 (0.9) | | |
| New confusion (%) | 61 (1.6) | 40 (1.6) | | |
| Pain (%) | 32 (0.8) | 17 (0.7) | | |
| Voice (%) | 151 (3.8) | 134 (5.3) | | |
| Unconscious (%) | 143 (3.6) | 63 (2.5) | | |
| Mean albumin (g/L) (SD) | 40.3 (5.7) | 40.2 (5.8) | 4484 | 0.508 |
| Mean creatinine (µmol/L) (SD) | 106.3 (104.1) | 103 (82.5) | 5125 | 0.194 |
| Mean haemoglobin (g/L) (SD) | 126.1 (23.4) | 127.5 (22.3) | 4680 | 0.027 |
| Mean potassium (mmol/L) (SD) | 4.4 (0.6) | 4.4 (0.6) | 4449 | 0.135 |
| Mean sodium (mmol/L) (SD) | 138.3 (5) | 137.9 (5.3) | 4349 | 0.016 |
| Mean white cell count ($10^9$ cells/L) (SD) | 10.3 (7.6) | 11 (5.9) | 5147 | <0.001 |
| Mean urea (mmol/L) (SD) | 7.9 (6.2) | 8.3 (6.6) | 4382 | 0.017 |
| AKI score | | | 2.2 | 0.158 |
| 0 (%) | 2900 (92) | 1916 (90.5) | | |
| 1 (%) | 137 (4.3) | 120 (5.7) | | |
| 2 (%) | 61 (1.9) | 46 (2.2) | | |
| 3 (%) | 53 (1.7) | 36 (1.7) | | |

AKI, Acute Kidney Injury; NEWS2, National Early Warning Score; SH, Scarborough Hospital; YH, York Hospital.

We report discrimination and calibration statistics as performance measures for these models.[13]

Discrimination relates to how well a model can separate—or discriminate between—those who died and those who did not and is given by the area under the receiver operating characteristics (ROC) curve (AUC) or c-statistic. The ROC curve is a plot of the sensitivity (true positive rate) versus 1−specificity (false positive rate) for consecutive predicted risks. A c-statistic of 0.5 is no better than tossing a coin, while a perfect model has a c-statistic

**Table 2** Performance of CARMc19_N and CARMc19_NB models for predicting the risk of mortality for patients with COVID-19 and patients without COVID-19 in validation dataset

| Model | COVID-19 | Mean risk discharged alive | Mean risk discharged deceased | ARD | Scaled Brier score | AUC (95% CI) | Calibration slope |
|---|---|---|---|---|---|---|---|
| CARMc19_N | No | 0.05 | 0.17 | 0.12 | 0.05 | 0.83 (0.79 to 0.86) | 1.11 (0.94 to 1.27) |
| CARMc19_N | Yes | 0.28 | 0.48 | 0.20 | 0.20 | 0.75 (0.69 to 0.81) | 0.85 (0.57 to 1.13) |
| CARMc19_N | All | 0.07 | 0.29 | 0.22 | 0.20 | 0.86 (0.83 to 0.88) | 0.95 (0.83 to 1.06) |
| CARMc19_NB | No | 0.05 | 0.20 | 0.15 | 0.10 | 0.87 (0.84 to 0.90) | 1.17 (0.99 to 1.35) |
| CARMc19_NB | Yes | 0.27 | 0.49 | 0.22 | 0.24 | 0.78 (0.71 to 0.84) | 0.93 (0.61 to 1.26) |
| CARMc19_NB | All | 0.07 | 0.30 | 0.23 | 0.22 | 0.88 (0.86 to 0.90) | 1.01 (0.88 to 1.14) |

.ARD, absolute risk difference; AUC, area under the curve; CARMc19, computer-aided risk score.

of 1. In general, values <0.7 are considered to show poor discrimination, values of 0.7–0.8 can be described as reasonable and values >0.8 suggest good discrimination.[11] The 95% CI for the c-statistic was derived using DeLong's method as implemented in the *pROC* library[12] in *R*.[14]

Calibration measures a model's ability to generate predictions that are, on average, close to the average observed outcome and can be readily seen on a scatter plot (y-axis=observed risk, x-axis=predicted risk). Perfect predictions should be on the 45° line. We internally validated and assessed the calibration for all the models using the bootstrapping approach.[15 16] The overall statistical performance was assessed using the scaled Brier score which incorporates both discrimination and calibration.[13] The Brier score is the squared difference between actual outcomes and predicted risk of death, scaled by the maximum Brier score such that the scaled Brier score ranges from 0% to 100%. Higher values indicate superior models.

The recommended threshold for detecting deteriorating patients and sepsis is NEWS2 ≥5.[17 18] Therefore, we assessed the sensitivity, specificity, positive and negative predictive values and likelihood ratios for these models at NEWS2 threshold of 4+, 5+ and 6+.[19] We followed the

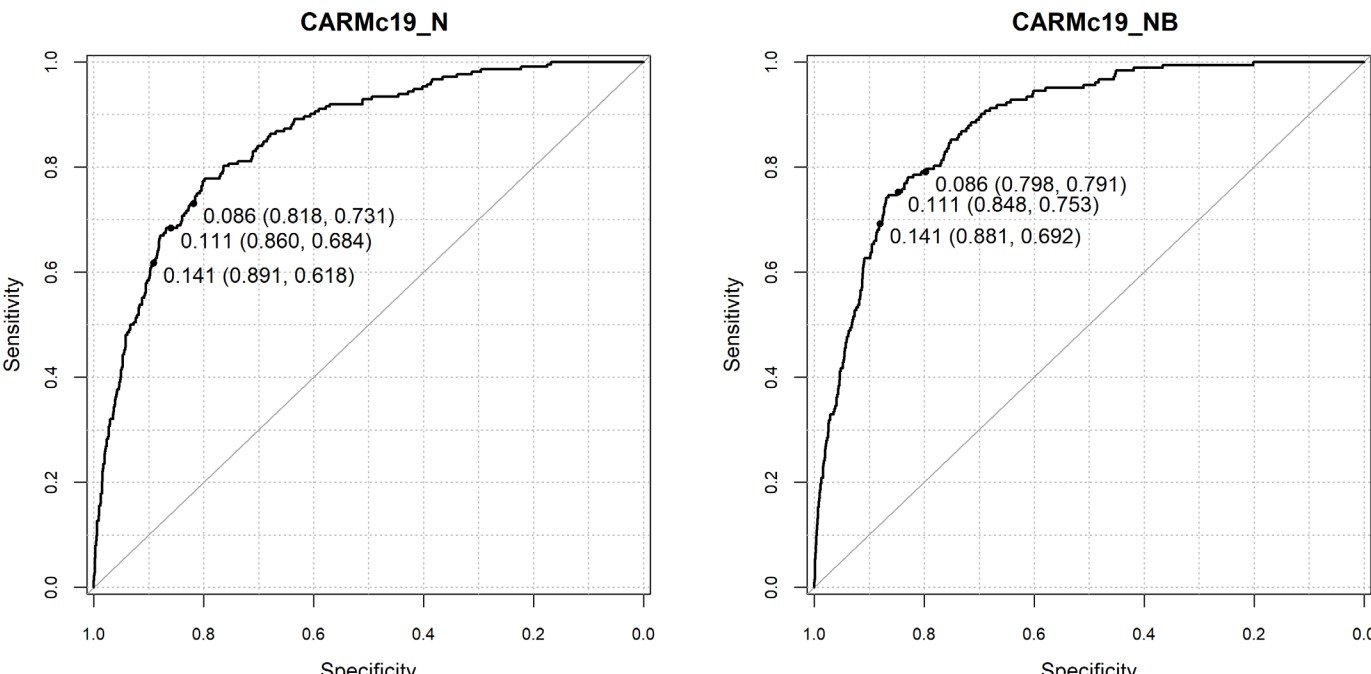

**Figure 1** Receiver operating characteristic curve for computer-aided risk score (CARMc19)_N and CARMc19_NB in predicting the risk of mortality in the development dataset. Predicted probability at National Early Warning Score thresholds 4+ (0.09), 5+ (0.11), 6+ (0.14) (sensitivity, specificity).

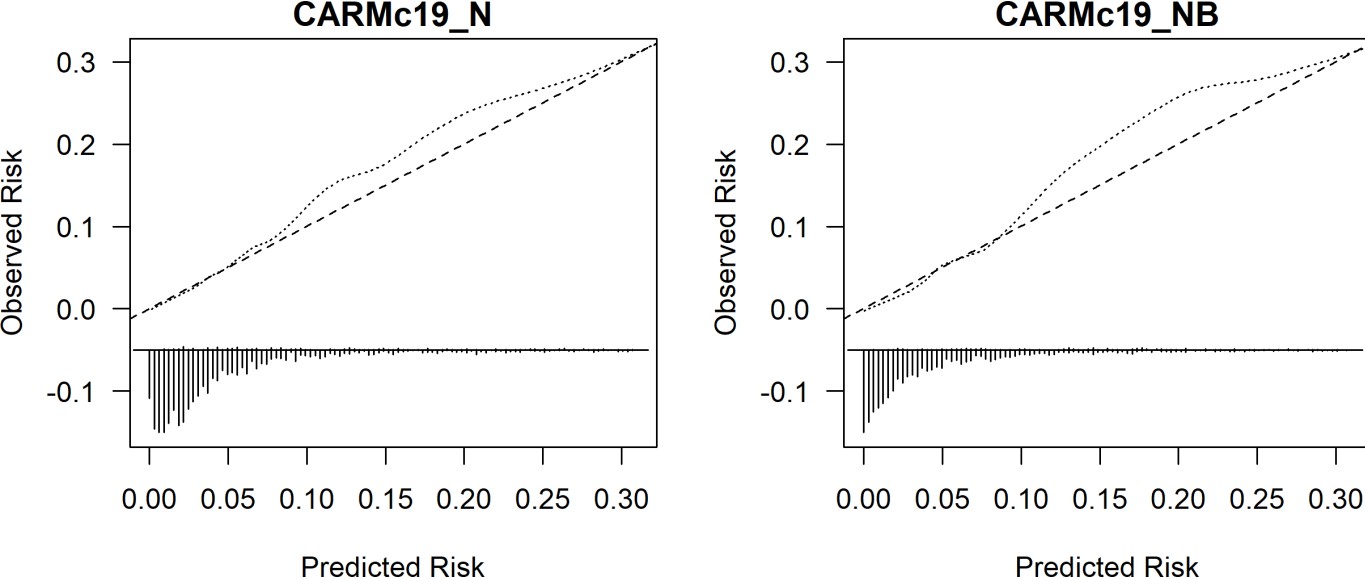

**Figure 2** External validation of computer-aided risk score (CARMc19)_N and CARMc19_NB models, respectively for predicting the risk of mortality. We limit the risk of mortality to 0.30 for visualisation purpose because beyond this point, we have few patients.

Transparent Reporting of a multivariable prediction model for Individual Prognosis Or Diagnosis (TRIPOD) guidelines for reporting of model development and validation.[20] We used Stata[10] for data cleaning and $R$[14] for statistical analysis.

## RESULTS
### Cohort characteristics
The number of non-elective discharges was 6444 over 3 months. For the development of CARMc19_N, we excluded 36 (0.6%) admissions because the index NEWS2 was not recorded within ±24 hours of the admission date/time, or these data were missing or not recorded at all (online supplemental appendix table S2). Likewise, for the development of CARMc19_NB, we further excluded 1189 (18.3%) of admissions because the first blood test results were not recorded within ±96 hours of the admission date/time, or they were missing or not recorded at all (online supplemental appendix table S2).

The characteristics of the admissions included in our study are shown in table 1. Emergency admissions in the validation dataset were older than those in development dataset (69.6 years vs 67.4 years), less likely to be male (49.5% vs 51.2%), had higher index NEWS2 (3.2 vs 2.8), higher prevalence of COVID-19 (11.0% vs 8.7%) but similar in-hospital mortality (8.4% vs 8.2%). See accompanying scatter plot and box plot in online supplemental appendix figure S1 to S4.

We assessed the performance of CARMc19_N and CARMc19_NB models to predict the risk of in-hospital mortality in emergency medical admissions (see table 2 and figure 1 for validation results and online supplemental appendix table S3 and figure S7 for model development results).

The c-statistics for predicting mortality for CARMc19_NB was slightly higher than model CARMc19_N in development dataset (CARMc19_NB=0.87 (95% CI 0.85 to 0.89) vs CARMc19_N=0.86 (95% CI 0.84 to 0.87)) and the validation dataset (CARMc19_NB=0.88 (95% CI 0.86 to 0.90) vs CARMc19_N=0.86 (95% CI 0.83 to 0.88)).

The c-statistics for predicting mortality for patients with COVID-19 lower than patients without COVID-19 (CARMc19_NB: 0.78 (95% CI 0.71 to 0.84) vs 0.87 (95% CI 0.84 to 0.90); CARMc19_N: 0.75 (95% CI 0.69 to 0.81) vs 0.83 (95% CI 0.79 to 0.86)).

Internal validation of both models is shown in online supplemental appendix figure S6. Both models had good internal and external calibration (CARMc19_NB: 1.01 (95% CI 0.88 vs 1.14) and CARMc19_N: 0.95 (95% CI 0.83 to 1.06)) (see table 2 and figure 2).

Table 3 includes the sensitivity, specificity, positive and negative predictive values for CARMc19_N and CARMc19_NB models for predicting mortality at NEWS2 threshold of 4+, 5+, 6+. At all NEWS2 thresholds (4+, 5+, 6+), model CARMc19_NB had better sensitivity (development dataset: 76% vs 72%; 71% vs 67%; 65% vs 61% and validation dataset: 79% vs 73%; 75% vs 68%; 69% vs 61%) and similar specificity (development dataset: 81% vs 82%; 86% vs 86%; 89% vs 90% and validation dataset: 80% vs 82%; 85% vs 86%; 88% vs 89%) (table 3 and online supplemental appendix table S4).

## DISCUSSION
In this study, we developed and validated two (CARMc19_N and CARMc19_NB) models to predict the risk of in-hospital mortality with the following covariates: (1) CARMc19_N uses age+sex+COVID-19 (yes/no)+NEWS2 including subcomponents; (2) CARMc19_NB extends

**Table 3** Sensitivity analysis of CARMc19_N and CARMc19_NB models in validation dataset for predicting the risk of mortality at NEWS2 thresholds 4+ (0.09), 5+ (0.11) and 6+ (0.14) of predicted risk of mortality in development dataset

| Model | At NEWS score (predicted risk of death) | Number of deaths identified by model | Sensitivity % | Specificity % | PPV | NPV | LR+ | LR– |
|---|---|---|---|---|---|---|---|---|
| CARMc19_N | 4+ (0.09) | 696 | 73.1 (66.6 to 79) | 81.8 (80.2 to 83.4) | 27 (23.4 to 30.8) | 97.1 (96.2 to 97.8) | 4 (3.6 to 4.5) | 0.3 (0.3 to 0.4) |
| CARMc19_N | 5+ (0.11) | 557 | 68.4 (61.7 to 74.6) | 86 (84.5 to 87.4) | 31 (26.8 to 35.4) | 96.7 (95.9 to 97.5) | 4.9 (4.3 to 5.6) | 0.4 (0.3 to 0.4) |
| CARMc19_N | 6+ (0.14) | 452 | 61.8 (54.9 to 68.4) | 89.1 (87.8 to 90.4) | 34.3 (29.5 to 39.3) | 96.2 (95.3 to 97) | 5.7 (4.9 to 6.7) | 0.4 (0.4 to 0.5) |
| CARMc19_NB | 4+ (0.09) | 651 | 79.1 (72.5 to 84.8) | 79.8 (77.9 to 81.6) | 26.9 (23.2 to 30.9) | 97.6 (96.7 to 98.3) | 3.9 (3.5 to 4.4) | 0.3 (0.2 to 0.3) |
| CARMc19_NB | 5+ (0.11) | 526 | 75.3 (68.3 to 81.4) | 84.8 (83.1 to 86.3) | 31.7 (27.3 to 36.3) | 97.3 (96.4 to 98) | 4.9 (4.3 to 5.6) | 0.3 (0.2 to 0.4) |
| CARMc19_NB | 6+ (0.14) | 431 | 69.2 (62 to 75.8) | 88.1 (86.5 to 89.5) | 35.3 (30.3 to 40.5) | 96.8 (95.9 to 97.6) | 5.8 (5 to 6.8) | 0.3 (0.3 to 0.4) |

CARMc19, computer-aided risk score; LR–, negative likelihood ratio; LR+, positive likelihood ratio; NPV, negative predictive value; PPV, positive predictive value.

model CARMc19_N with all seven blood test results and AKI score (online supplemental appendix figure S5). We found that CARMc19 scores have good performance chracterstiics and our findings tentatively suggest that a NEWS2 threshold of 5+ appears to strike a reasonable balance between sensitivity and specificity. CARMc19_NB was more sensitive with similar specificity than the CARMc19_N model.

CARMc19 scores performed better than our previous CARM models[6 7] because of additional NEWS2 variables (oxygen flow rate and oxygen scale 2) and COVID-19 status. A recent systematic review identified models to predict mortality from COVID-19 with c-statistics that ranged from 0.87 to 1.[21] However, despite these high c-statistics, the review authors cautioned against the use of these models in clinical practice because of the high risk of bias and poor reporting of studies which are likely to have led to optimistic results.[21] In contrast, our approach follows rigorous methodological standards for the development of risk scores.[22–24]

The main advantages of our models are that they are designed to incorporate data which are already available in the patient's electronic health record thus placing no additional data collection or computational burden on clinicians, and are readily automated. Nonetheless, we emphasise that our CARMc19 scores are not designed to replace clinical judgement. They are intended and designed to support, not subvert, the clinical decision-making process and can be always overridden by clinical concern.[5 25] The working hypothesis for our models is that they may enhance situational awareness of mortality by processing information already available without impeding the workflow of clinical staff, especially as our approach offers a faster and less expensive assessment of in-hospital mortality risk than current laboratory tests which may be more practical to use for large numbers of people.

There are limitations in relation to our study. We identified COVID-19 based on ICD-10 code 'U071', which was determined by COVID-19 swab test results (hospital or community) and clinical judgement and so our findings are constrained by the accuracy of these methods.[26 27] This does, however, allow the algorithm to take account of the entry of diagnostic information by the clinician including radiology findings as input variables if the swab result is negative. The systematically lower c-statistics for COVID-19 admissions requires further study. There are several candidate hypotheses which stem from the complex pathology of COVID-19—which can produce an inflammatory response (sepsis), coagulopathy (leading to sudden pulmonary embolism or arterial thrombosis). It is known that NEWS(2) is inadequate in monitoring hospital patients at risk of neurological deterioration, and this may also apply, to some extent, to COVID-19. Also, COVID-19 status could has a longer 'sell by date'. A PCR test may be positive up to 90 days after the initial infection and may therefore overestimate risk, if the patient is admitted and positive, when the COVID-19

episode is effectively over. Conversely, the physiological and pathological variables are unlikely to reflect the future risk if mortality is secondary to a sudden event such as veno-thromboembolism. COVID-19 diagnosis may also be determined by clinical diagnosis (as well as PCR positive test), whereas the other variables in our models are measurements (also subject to error, but less so than a diagnostic category).

We used the index NEWS2 data in our models, but vital signs and blood test results are repeatedly updated for each patient according to hospital protocols. Although we developed models using one hospital's data and validated into another hospital's data, the extent to which changes in vital signs over time reflect changes in mortality risk need to be incorporated in our models requires further study. Our two hospitals are part of the same NHS Trust and this may undermine the generalisability of our findings, which merit further external validation.

Although we focused on in-hospital mortality (because we aimed to aid clinical decision making in the hospital), the impact of this selection bias needs to be assessed by capturing out-of-hospital mortality by linking death certification data and hospital data. CARMc19, like other risk scores, can only be an aid to the decision-making process of clinical teams[11 28] and its usefulness in clinical practice remains to be seen.

The next phase of this work is to field test CARMc19 scores by carefully engineering it into routine clinical practice to see if it does enhance the quality of care for acutely ill patients, while noting any unintended consequences.

## CONCLUSION

We developed a validated a risk predictor (CARMc19 score) with good performance characteristics for predicting the risk of in-hospital mortality following an emergency medical admission during the pandemic where a significant proportion of the patient cohort was presenting with COVID-19 disease. Since the presentation of the CARMc19 scores to the clinician's caring for the patient placed no additional data collection burden on clinicians and is readily automated, it was carefully introduced to the electronic care record for clinicians caring for patients with COVID-19 in the hospital during the second phase of the pandemic.

## Author affiliations
[1]Faculty of Health Studies, University of Bradford, Bradford, UK
[2]Wolfson Centre for Applied Health Research, Bradford Royal Infirmary, Bradford, UK
[3]NIHR Yorkshire and Humber Patient Safety Translational Research Centre (YHPSTRC), Bradford, UK
[4]The Strategy Unit, NHS Midlands and Lancashire Commissioning Support Unit, West Bromwich, UK
[5]Department of Renal Medicine, York Teaching Hospital NHS Foundation Trust, York, UK
[6]Department of Information Technology, York Teaching Hospitals NHS Foundation Trust, York, UK

Contributors DR and MAM had the original idea for the work. KB, RH provided the data extracts. MF undertook the statistical analyses with support from MAM. MF, MAM, and DR wrote the first draft of the paper. DR provided clinical perspectives. All others contributed to the final paper and have approved the final version. DR & MF will act as study guarantors.

Funding This research was supported by the Health Foundation. The Health Foundation is an independent charity working to improve the quality of healthcare in the UK. This research was also supported by the National Institute for Health Research (NIHR) Yorkshire and Humber Patient Safety Translational Research Centre (NIHR Yorkshire and Humber PSTRC).

Disclaimer The views expressed in this article are those of the author(s) and not necessarily those of the NHS, the Health Foundation, the NIHR or the Department of Health.

Competing interests None declared.

Patient and public involvement Patients and/or the public were not involved in the design, or conduct, or reporting, or dissemination plans of this research.

Patient consent for publication Not applicable.

Ethics approval This study was deemed to be exempt from ethical approval because it was classified as an evaluation. Furthermore, this study used already de-identified data from an ongoing study involving NEWS, which received ethical approval from Health Research Authority (HRA) and Health and Care Research Wales (HCRW) (reference number 19/HRA/0548).

Provenance and peer review Not commissioned; externally peer reviewed.

Data availability statement Data may be obtained from a third party and are not publicly available. Our data sharing agreement is with York Hospital and does not permit us to share the data used in this paper.

ORCID iD
Muhammad Faisal http://orcid.org/0000-0003-4885-4251

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
