## [Reviewer comments · BMJ Open]

ARTICLE DETAILS

TITLE (PROVISIONAL)	Development and validation of automated computer aided-risk scores to predict in-hospital mortality for emergency medical admissions with COVID-19: a retrospective cohort development and validation study
AUTHORS	Faisal, Muhammad; Mohammed, Mohammed; Richardson, Donald; Fiori, Massimo; Beatson, Kevin

VERSION 1 – REVIEW

REVIEWER	Solis Garcia del Pozo, Julian Albacete University Hospital Complex
REVIEW RETURNED	28-Apr-2021

GENERAL COMMENTS	I thanks BMJ open to the opportunity to review this paper. The authors develop a computed aided-risk score to predict in-hospital mortality for hospital admissions, including COVID-19 patients. The authors performed the development of the score in York Hospital and the validation in Scarborough hospital. The prevalence of COVID-19 patients in the development dataset was 8.7%, and in the validation dataset, 11%. However, the title seems to reflect that the score has been developed exclusively for COVID-19 patients. The title should be improved to fit the content of the article better. On the other hand, I think that the methodology used is adequate, the manuscript is well written, and the information provided is relevant.
--

REVIEWER	Tulaimat, Aiman Cook County Health and Hospitals , Pulmonary, Critical Care, and Sleep Medicine
REVIEW RETURNED	10-May-2021

GENERAL COMMENTS	The authors attempt reexamine the CARM during the COVID pandemic. They have mastered this particular analysis technique. I do have few question to ask. Did they derive new coefficient for the various variables in the model? If yes, have they changed from prior models they have published? What does this change mean? What was the accuracy of the NEWS2? What is the threshold used to calculate the sensitivity and specificity for the CARM? Was it the same at each NEWS2? But the ultimate question is, how will this be implemented?
---

	The calculator integrated with the medical record will generate a number, but then what will be posted in the medical record? a text variable (low, intermediate, high)? Or a numerical probability of death? How frequently will the model need to be calibrated. You have tested your models many times now. How are they changing over time (connects with earlier point)? Death in the patients included in the analysis occurred despite medical care. Do the authors worry that clinicians will withhold effective treatments if the model predicts death? Did any of the patients have a terminal illness included? Is so, can the analysis rerepeated without them? Consider also removing all the patient that died within 2-3 days of admission. These might have clear prognosis on admission and a model is not needed. How does the model preform if the patients with extreme values are removed. Examples of extreme values are respiratory rate > 35, heart rate > 120, SBP < 80, etc. These are situations that are easy to identify by the clinician and a model is not needed to detect severity of illness. You can calculate the c statistic for patients above and below each of these value. You can also calculate the c for patients with no extreme values and patients with any. How did the model perform in COVID patients?
--	---

VERSION 1 – AUTHOR RESPONSE

Reviewer: 1

Dr. Julian Solis Garcia del Pozo, Albacete University Hospital Complex Comments to the Author:
I thanks BMJ open to the oppportunity to review this paper. The authors develop a computed aided-risk score to predict in-hospital mortality for hospital admissions, including COVID-19 patients.

The authors performed the development of the score in York Hospital and the validation in Scarborough hospital. The prevalence of COVID-19 patients in the development dataset was 8.7%, and in the validation dataset, 11%. However, the title seems to reflect that the score has been developed exclusively for COVID-19 patients. The title should be improved to fit the content of the article better.

On the other hand, I think that the methodology used is adequate, the manuscript is well written, and the information provided is relevant.

Response: Development and validation of automated computer aided-risk scores to predict in-hospital mortality for emergency medical admissions with COVID-19: a retrospective cohort development and validation study

Reviewer: 2

Dr. Aiman Tulaimat, Cook County Health and Hospitals Comments to the Author:

The authors attempt reexamine the CARM during the COVID pandemic.

They have mastered this particular analysis technique.

I do have few question to ask.

Did they derive new coefficient for the various variables in the model?
 If yes, have they changed from prior models they have published? What does this change mean?
 What was the accuracy of the NEWS2?

Response: There are two major aims of statistical modelling: 1) causal inference (focussed on individual variable impact by inspecting the regression coefficients) and 2) predictive modelling (focused on the predictive performance of outcome in terms of discriminating the adverse outcome patients). In this study, we adopt a later approach and aimed to develop a model (CARMc19_N/NB) for predicting in-hospital mortality by updating the previous version (CARM_N/NB) for COVID-19 patients using NEWS2 data. We found the CARMc19 models are better than previous models in terms of discrimination (c-statistic or AUC).

	NEWS (_N)	NEWS + Blood (_NB)
Updated models CARMc19	0.86 (0.83 to 0.88)	0.88 (0.86 to 0.90)
Previous models CARM	0.82 (0.81 to 0.83)	0.86 (0.85 to 0.87)

What is the threshold used to calculate the sensitivity and specificity for the CARM? Was it the same at each NEWS2?

Response: We calculated the predicted risk of death from a simpler logistic regression model (died~NEWS2) at three NEWS2 cut-offs (4, 5, 6). We used these predicted risk of death probabilities as thresholds for calculating the sensitivity and specificity.

But the ultimate question is, how will this be implemented?

Response: CARMc19 is currently at the implementation stage. Please see the screenshot below that show how its implementation will look like. We nevertheless agree that sound implementation and rigorous evaluation remain key considerations.

The calculator integrated with the medical record will generate a number, but then what will be posted in the medical record? a text variable (low, intermediate, high)? Or a numerical probability of death?
 Response: A numerical probability of death (from 0 to 1) will be posted in the medical record based on insights from our qualitative research work (Dyson et al., 2019).

Dyson J., March C., Faisal M., Richardson D., Scally A., Beatson K., Howes R., Speed K., Jackson N., Mohammed M.A. (2019). Understanding and applying practitioner and patient views on the implementation of a novel automated Computer Aided Risk Score (CARS) predicting the risk of death following emergency medical admission to hospital: A qualitative study. *BMJ Open*. DOI: 10.1136/bmjopen-2018-026591

How frequently will the model need to be calibrated?

Response: Our aim is to assess the model predictive performance (in terms of discrimination and calibration) annually or if there is any indication from medical staff that they feel the scores lack face validity.

You have tested your models many times now. How are they changing over time (connects with earlier point)?

Response: The model predictive performance (in terms of discrimination and calibration) has been improved with the addition of COVID-19 status.

Death in the patients included in the analysis occurred despite medical care. Do the authors worry that clinicians will withhold effective treatments if the model predicts death?

Response: CARM is comparable with medical judgements in discriminating in-hospital mortality following emergency admission to an elderly care ward (Faisal et al., 2019). CARM may have a promising role in supporting medical judgements in determining the patient's risk of death in the hospital. It is important to note that we have designed CARM to support the medical decision-making process, not replace it, without placing any additional data collection burden on staff. We nevertheless agree that such questions are very important and need further research.

Faisal M., Khatoon B., Scally A., Richardson D., Irwin S., Davidson R., Heseltine D., Corlett A., Ali J., Hampson R., Kesavan S., McGonigal G., Goodman K., Harkness M., Mohammed M.A. (2019). A prospective study of consecutive emergency medical admissions to compare a novel automated computer aided mortality risk score and clinical judgement of patient mortality risk. *BMJ Open* DOI: 10.1136/bmjopen-2018-027741

Did any of the patients have a terminal illness included? Is so, can the analysis re-repeated without them?

Response: Unfortunately we don't know who might have had a terminal illness. However, Patients die not from their disease but from the disordered physiology caused by the disease. [McGinley & Pearse. A national early warning score for acutely ill patients. *BMJ* 2012;345:e5310 doi: 10.1136/bmj.e5310 (Published 8 August 2012)]. So our algorithm is designed to predict who is at risk of death in this current episode in the hospital using the variables listed. Many patients may have a 'terminal' illness but survive for months/years. Heart failure, end-stage renal failure, metastatic prostate/breast cancer, multiple sclerosis etc. Diagnosis and Comorbidities were not part of the risk equation.

Consider also removing all the patient that died within 2-3 days of admission. These might have clear prognosis on admission and a model is not needed.

Response: We have now compared the performance of models before and after removing the patient that died within 2 days of admission

	NEWS (_N)	NEWS + Blood (_NB)
After removing all the patient that died within 2 days of admission	0.85 (0.84 to 0.86)	0.88 (0.86 to 0.90)
Before removing all the patient that died within 2 days of admission	0.86 (0.84 to 0.88)	0.88 (0.86 to 0.90)

How does the model preform if the patients with extreme values are removed. Examples of extreme values are respiratory rate > 35, heart rate > 120, SBP < 80, etc. These are situations that are easy to identify by the clinician and a model is not needed to detect severity of illness. You can calculate the c

statistic for patients above and below each of these value. You can also calculate the c for patients with no extreme values and patients with any.

Response: We have now compared the performance of models in the groups as suggested.

	NEWS (_N)	NEWS + Blood (_NB)
Respiratory rate =< 35	0.84 (0.82 to 0.87)	0.87 (0.85 to 0.90)
Respiratory rate > 35	0.89 (0.83 to 0.97)	0.87 (0.79 to 0.96)
Pulse rate =< 120	0.85 (0.83 to 0.88)	0.88 (0.86 to 0.90)
Pulse rate > 120	0.88 (0.80 to 0.96)	0.86 (0.77 to 0.95)
Systolic blood pressure => 80	0.85 (0.83 to 0.88)	0.88 (0.86 to 0.90)
Systolic blood pressure < 80	0.96 (0.93 to 0.99)	0.94 (0.90 to 0.98)
With any of above extreme values	0.88 (0.86 to 0.90)	0.88 (0.85 to 0.90)
None of above extreme value	0.87 (0.79 to 0.95)	0.90 (0.85 to 0.95)

How did the model perform in COVID patients?

Response: We have now reported the performance of models in COVID-19 and non-COVID-19 patients in the results section of the paper. Furthermore, we also added less accurate predictions for COVID-19 patients as a limitation in the discussion section.

	NEWS (_N)	NEWS + Blood (_NB)
Non_COVID-19	0.83 (0.79 to 0.86)	0.87 (0.84 to 0.90)
COVID-19	0.75 (0.69 to 0.81)	0.78 (0.71 to 0.84)
All	0.86 (0.83 to 0.88)	0.88 (0.86 to 0.90)

VERSION 2 – REVIEW

REVIEWER	Solis Garcia del Pozo, Julian Albacete University Hospital Complex
REVIEW RETURNED	24-Oct-2021

GENERAL COMMENTS	From my point of view, the authors have answered the reviewers' questions and made appropriate changes to the manuscript. I think the article can be published in its current version.
--